# Nanomaterials for Wound Healing and Infection Control

**DOI:** 10.3390/ma12132176

**Published:** 2019-07-06

**Authors:** Mara Madalina Mihai, Monica Beatrice Dima, Bogdan Dima, Alina Maria Holban

**Affiliations:** 1Dermavenereology Department, Emergency University Hospital “Elias”, 011461 Bucharest, Romania; 2Department of Oncologic Dermatology-Emergency University Hospital “Elias”, University of Medicine and Pharmacy “Carol Davila”, 020021 Bucharest, Romania; 3Department of Microbiology and Immunology, Faculty of Biology, University of Bucharest, 030018 Bucharest, Romania; 4Department of Science and Engineering of Oxide Materials and Nanomaterials, Faculty of Applied Chemistry and Materials Science, University Politehnica of Bucharest, 1–7 Polizu Street, 011061 Bucharest, Romania

**Keywords:** wound healing, infections control, chronic infection, nanoparticles, nano-dressings

## Abstract

Wound healing has been intensely studied in order to develop an “ideal” technique that achieves expeditious recovery and reduces scarring to the minimum, thus ensuring function preservation. The classic approach to wound management is represented by topical treatments, such as antibacterial or colloidal agents, in order to prevent infection and promote a proper wound-healing process. Nanotechnology studies submicroscopic particles (maximum diameter of 100 nm), as well as correlated phenomena. Metal nanoparticles (e.g., silver, gold, zinc) are increasingly being used in dermatology, due to their beneficial effect on accelerating wound healing, as well as treating and preventing bacterial infections. Other benefits include: ease of use, less frequent dressing changes and a constantly moist wound environment. This review highlights recent findings regarding nanoparticle application in wound management.

## 1. Introduction

Wound healing involves three biological stages: inflammation, which usually lasts up to six days, proliferation, which typically covers the following two weeks, and remodeling which continues for up to two years. It is important to mention that in vivo the phases overlap due to several anatomical and physiological factors such as the constant intercellular signaling that implies real-time feedback control of pro-inflammatory and anti-inflammatory cytokine release. The inflammatory phase begins with a vascular response that ensures hemostasis through the formation of blood clots. Secondarily, distress molecules released by injured cells act as chemoattractants for leukocytes which exhibit two main functions: recognition and destruction of infectious agents, as well as cytokine release that stimulates cells involved in the proliferative phase [1]. The moment when granulation tissue begins to cover the wound surface marks the transition to the proliferative phase. Key factors in this second stage are represented by activation of fibroblasts which produce collagen and other extracellular matrices, as well as by neoangiogenesis [2]. The third phase, of remodeling restores the morphology and the function of the tissue [1].

The physiology of the healing process may be perturbed by both external and internal factors such as necrotic tissue, contamination with pathogens, foreign material entrapment, as well as underlying comorbidities (e.g., diabetes mellitus). All of these factors may prolong each phase, leading to a structurally and functionally unsatisfactory result [2].

The most frequently encountered issue in wound closure is the colonization with an infectious agent. Even though the bacteria that form the skin microbiota have a beneficial role in preventing the colonization of other pathogens, once they reach a critical threshold at the site of a wound, they can impede the healing process, especially if they form biofilms. In addition, external pathogens can directly infect the wound, leading to an inadequate healing process [2]. Out of these, the most commonly detected are *Staphylococcus aureus*, methicillin-resistant *Staphyloccoccus aureus* (MRSA), which affect the initial phases of wound healing, whereas *Pseudomonas aeruginosa* and *Escherichia coli* are characteristic for chronic wounds and they infect deeper skin layers [3].

Chronic wound infections are commonly polymicrobial, due to synergistic bacterial development patterns [4]. For example, aerobic bacteria require more oxygen in order to support their cell division, thus leading to a hypoxic tissue environment. In turn, this favors the development of anaerobic bacteria which produce short chain fatty acids that are capable of blocking the phagocytic action of host defense cells (Figure 1). As a result, even more pathogens can colonize the wound. Moreover, different types of microorganisms can exchange nutrients, thus sustaining each other’s proliferation. This explains why, in some chronic wounds, *Staphylococcus aureus* and *Pseudomonas aeruginosa* grow simultaneously in co-cultures [2].

Bacterial biofilms, mostly described in chronic wounds, represent an enclosed environment created by the colonizing bacteria through the production of an extracellular polymer which acts as a protective layer [3,4,5]. Biofilms generate a reactive oxygen dependent state of chronic inflammation, which inhibits re-epithelialization. Moreover, biofilms protect the bacterial colonies against the host immune response, as well as antibacterial therapies [6]. Biofilms can also be polymicrobial, the most common aerobic bacterial genus observed being *Staphylococcus* and *Enterococcus*, whereas the most common anaerobics were *Prevotella* and *Prophorymonas* [3,4,5].

Current wound healing therapies generally fail to provide a good clinical outcome, either structurally (e.g., wound re-epithelialization, control of fluid loss), or functionally (e.g., histological features that determine elasticity, durability, sensitivity, etc.). For this reason, nanotechnology, through the versatile physicochemical properties, is a reliable research domain for wound-healing therapies. By varying the type of material, the size and electrical charge of the nanoparticles, their biochemical features such as hydrophobicity, interaction with biological targets and deeper levels of tissue penetration can easily be adjusted for potentially any type of wound [7].

Acute wounds are usually characterized by a reduced microbial load/no infection, scab formation and immune cells infiltration in the initial phase, while healing is correlated with re-epithelialization, angiogenesis and fibroblast migration. If the wound in critically colonized a localized infection is usually installed. This is characterized by the formation of microbial microcolonies and the release of some biochemical molecules, such as quorum sensing (QS) signals. If the immune system is not able to clear the infection, microbial biofilms are allowed to develop and the wound becomes persistent. Chronic infection is characterized by an increased inflammatory process, lower oxygenation of the deep tissues due to the formation of fibrin cuffs, fibroblast senescence, impaired angiogenesis and stalled re-epithelialization. 

## 2. Current Wound Therapy

Wound dressings have been traditionally applied in order to compensate for the lack of a protective barrier at the site of the wound, thus shielding it from external contamination. Additional functionality has been achieved through the association of different therapeutic complexes. Therefore, classic wound dressing materials (e.g., cotton and wool) which only act as a passive barrier, are being replaced by more technologically advanced dressings that are capable of creating a protective environment, while also allowing the transfer of active compounds. As a result, different combinations of both synthetic and natural materials have been developed: sponges, hydrogels, films, hydrocolloids, hydrofiber mats etc. [2].

The ideal dressing must ensure an environment which is maintained moist, while also avoiding the accumulation of wound exudates. In addition, through the control of local temperature, it also stimulates proper tissue perfusion. Therefore, key characteristics must include: biocompatibility, semi-permeability to water and oxygen, as well as being hypoallergenic. Moreover, it must promote tissue renewal processes and must not be traumatic upon removal. Nevertheless, it must also be cost effective [2].

Dressings that ensure a moist environment, suitable for fast re-epithelialization (e.g., hydrocolloid, semi-permeable polyurethane, calcium alginate) are recommended for sterile wounds [8]. However, the management of necrotic or infected wounds requires the usage of ‘functionalized’ dressings embedded with different classes of antibiotics or other antibacterial substances [2].

The most used antibiotics in antimicrobial wound dressings are quinolones, tetracyclines, aminoglycosides and cephalosporins. They can block bacterial proliferation through different mechanisms: alteration of protein and nucleic acid syntheses which can lead to metabolic imbalances and the disruption of bacterial cell wall integrity thus also impairing cellular division [2]. However, the repeated and improper administration of antibiotics can determine the selection of resistant bacterial colonies—approximately 70% of bacteria that colonize chronic wounds present antibiotic resistance to at least one commonly prescribed agent [9].

Taking into account the compromised blood circulation in most chronic wounds, oral antibiotics may be ineffective due to low reachability to damaged tissues. Oral or topical antibiotics are not recommended for the treatment of chronic wounds due to the development of resistance and tolerance of bacteria. Topical non-antibiotic antimicrobials are preferred, considering the ease of application and rarity of systemic toxicity [10,11,12].

In order to prevent bacterial resistance to antibiotics, unconventional, non-antibiotic treatments, such as essential oils and honey, are frequently associated with wound dressings [13]. Essential oils are secondary plant metabolites that exhibit regenerative properties through the reduction of oxidative stress, inflammation, as well as viral and bacterial proliferation. They destroy the bacterial phospholipid cell membrane thus disrupting cellular compartments through pH imbalance and cytoplasm outburst. Due to the fact that essential oils do not cause antimicrobial resistance, they can be used as treatment for multidrug resilient microorganisms [14]. However, repetitive applications may result in adverse reactions, such as contact dermatitis.

Honey has been observed to promote granulation and re-epithelialization through angiogenesis and anti-inflammatory properties. In addition, its acidic pH stimulates macrophages to phagocyte bacteria and inhibit the development of microbial biofilm. Moreover, honey contains hydrogen peroxide, lysozyme, phenolic acids, which trigger an oxidative cascade damaging for pathogens [2]. In order to improve the efficiency of such natural products and decrease the potential associated side effects, authors have proposed the use of nanostructured materials as stabilizing agents and shuttles to ensure a controlled release and targeted delivery of natural and synthetic antimicrobials [15].

Hydrogels consist of a polymeric network with numerous hydrophilic groups that cross-interact, forming a three dimensional matrix which traps liquids, such as water or wound exudates [10]. Hydrogel-based wound dressings overcome the limitations of classic dressings by creating a moist environment which also permits gas diffusion. Moreover, by absorbing wound exudates, they considerably lower the risk of dressing-dependent infections. Different types of materials have been employed as wound dressings, such as: synthetic polymers (e.g., polyvinyl alcohol) or natural polymers (e.g., alginate, chitin, chitosan) [16], some of them with intrinsic antimicrobial properties (i.e., chitosan based materials).

Silver products (e.g., silver nitrate, silver sulfadiazine) are commonly used in infected chronic wounds and burns dressings, due to the release of silver ions which exhibit a potent antibacterial effect. Silver ions bind to thiol groups of peptidoglycans, causing bacterial cell lysis [17]. In addition, microbial DNA is altered through the blockage of respiratory enzyme pathways. Moreover, silver compounds are effective against multi-resistant bacteria, as well as bacterial biofilms. However, silver-derived products may cause tissue toxicity [7].

Even if it has been demonstrated that chronic wounds are characterized by a local depletion of growth factors, topical application without a proper dispensing vector does not influence re-epithelialization. However, only Rh-PDGF (recombinant human platelet-derived growth factor) has been licensed for treating wounds. PDGF dimers (PDGF-AA, PDGF-BB or PDGF-AB), composed of two polypeptide chains (A or B), are released by different cells involved in the stages of wound healing, such as: endothelial cells, thrombocytes, fibroblasts, as well as macrophages, monocytes and keratinocytes. Studies have shown that, when integrated in a gel (i.e., becaplermin gel 0.01%, 100 µg/g), topical recombinant human homodimer PDGF-BB stimulates granulation in chronic diabetic foot ulceration, without any major adverse effects [8].

Several clinical trials performed on rodents have discovered that an increase in TGFβ-3 isoform (transforming growth factor β-3) led to a proper dermal restoration, resulting in higher tensile strength, without an increase in healing time or excessive scarring. The modulation of the TGFβ isoform ratio in favor of TGFβ-3 was achieved through direct administration of recombinant TGFβ-3 or through antibodies targeted against TGFβ-1 and -2 isoforms [8].

Another target in wound healing regulation is the serine protease-activated protein-C (APC) which reduces inflammation and promotes angiogenesis and re-epithelialization through the expression of a variety of mediators such as MMP-2 (*matrix metalloproteinase*-2), VEGF (vascular endothelial growth factor) and MCP-1 (*membrane cofactor protein*-1) in keratinocytes, endothelial cells and fibroblasts [8].

VAC (vacuum-assisted closure) Therapy is another important adjuvant in wound healing which employs negative pressure driven suction. As a result, it reduces tissue edema and also applies micromechanical stretching forces to individual cells, causing them to express transcription factors, thus promoting cell proliferation and angiogenesis. Moreover, it can stimulate the production of healthy granulation tissue after surgical debridement of previously infected wounds [8].

Several bioengineered skin substitutes, targeting the dermal injury, have been developed. Acellular networks stimulate matrix synthesis in order to restore dermal architecture. Alloderm© is an allogenic, cadaveric human skin that is applied on the wound bed, allowing skin autografts to be properly integrated. Integra© consists of a two-layer system: deep layer of bovine collagen rich in chondroitin sulphate, and a silicone sheet, which covers the area. Once it adheres to the wound, the silicone is replaced with skin grafts, which do not contract as much as directly applied grafts. Other options use transplanted cells embedded into dermal matrices that can accelerate healing. For example, Dermagraft-TC© is a nylon mesh with integrated fibroblasts, whereas Apligraf© employs both allogenic neonatal keratinocytes as well as a dermal matrix composed of bovine type 1 collagen and fibroblasts. Moreover, if autologous keratinocytes are cultured and integrated into the Integra dressing, they are able to migrate superficially and aid the epidermal restoration [8]. More clinical studies need to be conducted in order to determine their advantages.

## 3. Nanomaterials in Wound Healing

### 3.1. Definition and Classification

Nanotechnology studies the synthesis, structure and dynamic of atomic and molecular nanometric particles (maximum diameter of 100 nm), termed nanoparticles, upon which nanoproducts are built [17]. By downscaling to nanometric range, a particle’s surface increases exponentially, whereas its volume decreases, which leads to particular physico-chemical characteristics that account for numerous medical applications. Dynamic light scattering (DLS), fluorescence correlation spectroscopy (FCS) and Raman scattering (RS) offer information regarding the hydrodynamic size distribution of nanoparticles, whereas circular dichroism (CD), Infrared spectroscopy (IR) and Mass spectroscopy (MS) help in investigating their structure, conformation and surface characteristics. In addition, electron microscopy in general and transmission electron microscopy (TEM) and scanning electron microscopy (SEM) in particular, can achieve resolutions of less than 1 nm, thus making them valuable investigational tools employed in studying the properties and dynamics (i.e., diffusion, adherence) of nanomaterials. Through TEM, the morphology of nanomaterials can be investigated, whereas SEM recreates a superficial image of the nanomaterial, thus being solely employed in the study of dimensions and overall shape [18].

The size and shape of nanoparticles are key properties that determine their biological efficiency, by influencing active substance delivery (carrier circulation), penetrability (either directly through cell membranes or through phagocytosis) and cellular responses (receptor recognition) [18,19,20].

Due to their antibacterial properties and low toxicity profile, metal nanoparticles such as silver, gold and zinc represent ideal candidates for integration in wound dressings [2]. The main types of nanomaterials used for wound treatment are represented by nanoparticles, nanocomposites, coatings and scaffolds (Figure 2).

Nanomaterials for tissue regeneration can be developed under different structures: nanoparticles, nanospheres, nanocapsules, nanoemulsions, nanocarriers and nanocolloids [17].

Two principal types of nanoparticles (NPs) are commonly employed in wound therapy: (1) NPs that possess intrinsic characteristics that aid wound closure; (2) NPs used as delivery vectors for therapeutic agents. The former can be divided into metallic/metal oxide nanomaterials and nonmetallic nanomaterials (Figure 2) [7].

Regarding metallic and metal oxide nanoparticles, the most studied are silver, gold and zinc compounds due to their unique properties: antibacterial activity and reduced penetration into the skin. Their effectiveness, as well as toxicity, relies on key features, for example: dimension and architecture (smaller particles are more biologically active), surface functionalization, zeta potential and polydispersity index [15]. Zeta potential estimates the particle surface charge which influences receptor binding capacity as well as the penetrability of cellular barriers [19]. Other particle surface characteristics, such as porosity, chemical structure and heterogeneity, hydrolytic stability, determine the biological behavior of nanomaterials, including their interactions with other biomolecules, which may influence their biodistribution [20,21].

Nanospheres are small polymeric matrix systems composed of a fixed porous polymer onto which active substances (e.g., aminoacids, minerals or organic substances) can bind. As a result, active compounds become more stable and exhibit increased biocompatibility and improved pharmaceutical properties [17].

Nanocapsules are capable of enclosing an active agent within their structure, releasing it at a specific time, ensuring a more controlled delivery. They can also improve the penetration of active ingredients into deeper layers of the dermis. On the other hand, the effectiveness of the substances may be reduced [17].

Nanoemulsions represent homogenous, thermally stable oil-in-water emulsions with a droplet size of maximum 100 nm that also contain surfactant and can easily incorporate biologically active substances, preventing their sedimentation. They are also characterized by high solubilization capacity and fluidity and decreased viscosity. Their disadvantage would be the uncontrolled accumulation of active substances in reticular dermis or subcutaneous fat [17].

Nanocarriers structurally resemble liposomes, but contain synthetic, non-ionic surfactants. In addition, they express surface receptors capable of binding to specific loci, thus increasing the efficacy of the active agent, while also minimizing any adverse reactions [17].

Nanocolloids are composed of 1.5 to 5 nm non-ionic metal nanoparticles highly scattered in demineralized water due to their perpetual Brownian motion which allows them to penetrate eukaryotic and prokaryotic cells, thus exhibiting antimicrobial activity. Moreover, they are characterized by good electrical conductivity, being positively charged particles [17].

### 3.2. Nanoparticles

#### 3.2.1. Silver Nanoparticles

Silver nanoparticles (AgNPs) might overcome the limitations of standard silver compounds. Due to their increased surface-to-volume ratio, AgNPs are more potent at reduced concentrations, thus lowering their toxicity. Pure silver nanoparticles can modulate anti-inflammatory cytokine release, in order to promote rapid wound closure without increasing scarring [7]. By inducing the differentiation of myofibroblasts from normal fibroblasts, AgNPs promote wound contractility, thus accelerating the healing process [21]. Moreover, AgNPs stimulate epidermal re-epithelialization through the proliferation and relocation of keratinocytes [21]. However, Szmyd et al. reported that higher concentrations of AgNPs decrease keratinocyte viability, metabolism, as well as migration and differentiation of these cells, through the activation of caspase 3 and 7 (proteases involved in programmed cell death) and dose-dependent DNA damage [22]. In order to reduce side effects, silver nanoparticles can be used in low doses along with antimicrobial drugs to achieve increased efficiency. In a recent study it was demonstrated that AgNPs combined with tetracycline significantly decreased bacterial load both in superficial and deep tissue layers in a mouse model, thus accelerating healing [23]. These results encourage the association of AgNPs with traditional antibacterial agents or dressings, in order to more efficiently treat infected wounds. For example, biocellulose functionalized with AgNPs acted as an antimicrobial coating for open wounds. This nanomaterial proved a high bacteria-killing performance against Gram negative pathogens and promoted wound healing [24]. Similar results have been reported by Holban et al. who coated polyester-nylon dressings with AgNPs which interfered with bacterial colonization as well as biofilm synthesis, while maintaining a low toxicity profile [25].

The main anti-bacterial mechanism of AgNPs is creating sulfuric bonds with either bacterial cell membrane proteins or thiol groups of various enzymes, especially those involved in the respiratory chain, thus leading to apoptosis [21]. Moreover, since DNA contains sulfurous and phosphorous bonds, AgNPs can interfere with DNA synthesis during cell division, thus inhibiting bacterial multiplication [21].

Lu et al. observed in animal trials that inorganic particles such as silica firmly adhere to open wounds. As a result, they integrated AgNPs into mesoporous silica nanoparticles through the aid of disulfide bonds (Ag-MSNs), the new compound demonstrating excellent antibacterial activities, with little cellular toxicity [26].

In a recent study performed in vivo on dogs, we have demonstrated that silver impregnated coatings ensures faster wound healing and avoid microbial colonization on site [6].

Moreover, silver nanoparticles were used for numerous clinical trials in the therapy of wounds, especially burns and chronic wounds (i.e., diabetic ulcers) [27]. Currently, there are some commercially available dressings containing AgNPs. Such example is Acticoat©-wound dressing containing AgNPs with average size of 15 nm. The proved properties of Anticoat© are: wound healing, infection reduction on the site of the wound, and pain reduction, aspects observed in most tested patients. However, this coating is still under investigation for the therapy of burns. Recent clinical trials concluded that Anticoat© may be efficient in avoiding infections in burns when applied together with silver sulphadiazine and chlorhexidine digluconate cream [28].

#### 3.2.2. Gold Nanoparticles

Gold nanoparticles (AuNPs) represent a solid option when it comes to wound therapy due to their chemical stability and capacity of absorbing near infrared light, while, at the same time, being relatively easy to synthetize [17]. Moreover, by tuning the surface plasmon resonance, AuNP gels gain thermoresponsiveness, as described by Arafa et al., who demonstrated their antibacterial and healing properties both in vitro and in vivo, backed by histopathologic examinations [10]. AuNPs can either directly target the bacterial cell wall, or they can bind to bacterial DNA, blocking the double-helix from uncoiling during replication or transcription, thus exerting bactericidal and bacteriostatic properties. As a result, they can inhibit multidrug-resistant pathogens, such as *Staphylococcus aureus* and *Pseudomonas aeruginosa*. Moreover, AuNPs prevent the formation of reactive oxygen species, thus acting as antioxidants, aiding the healing process [21].

Lu et al. reported that while low concentrations of AuNPs enhance keratinocyte growth and differentiation, higher levels were associated with cytotoxicity [29]. Based on the observation that bioactive glass induces local cell response and the release of bFGF (basic fibroblast growth factor) and VEGF (vascular endothelium growth factor), Marza et al. studied the effect on wound healing of bFGF combined with AuNPs in Vaseline mixtures at 6, 12 and 18 wt% for 14 days. The results showed that the compound stimulated angiogenesis and fibroblast proliferation, promoting rapid wound closure, especially in the 18 wt% composite group, without causing cell toxicity [30]. AuNPs can be efficient in the therapy of burns also, being able to promote healing and inhibit microbial colonization, while being transdermally active, as demonstrated by a recent ex vivo permeation study [10].

#### 3.2.3. Zinc Oxide Nanoparticles

Zinc oxide nanoparticles (ZnONPs) represent a reliable antibacterial agent, by inducing bacterial cell membrane perforations. Moreover, when incorporated in hydrogel-based wound dressings [7], the overall contact time is increased, promoting keratinocyte migration, thus improving re-epithelialization [21]. In addition, in a recent study, a microporous chitosan hydrogel/ZnONPs dressing presented a high capacity of absorbing wound exudates and enabled the formation of hemostatic blood clots, while also displaying antibacterial properties with little cytotoxicity [16].

Aiming to provide a better alternative for treating burns while also reducing the incidence of associated sepsis, Balaure et al. described the synthesis and characteristics of dressings that incorporated ZnONPs in a bioresorbable matrix composed of collagen and 1% orange essential oil. The wound dressing accelerated wound closure and prevented bacterial growth both in vitro and in vivo, while also displaying excellent biocompatibility and reduced cytotoxicity [31].

However, the intrinsic high toxicity of these material limit their usage in wound healing therapies [7]. For example, high concentrations of ZnONPs are associated with mitochondrial dysfunction in keratinocytes, causing the release of lactate dehydrogenase. Moreover, ZnONPs produce reactive oxygen species, while also inhibiting the expression of superoxide dismutase and glutathione peroxidase genes in human keratinocytes, inducing cell membrane oxidative stress and apoptosis. In addition, it has been described that ZnONPs generate carcinogenic transformations [32].

Researchers have also tried to create core-shell nanocomposites by combining two metals. Khan et al. covered biogenic AuNPs extracted from *Hibiscus sabdariffa* plant with a thin layer of ZnO, synthetizing AuZnO core-shell nanocomposites, which exhibited antibacterial and anti-biofilm effects against *Staphylococcus aureus* and methicillin resistant *Staphylococcus haemolyticus*, through the release of reactive oxygen species [33]. In a skin wound mouse model, ZnONPs achieve successful wound closure and aesthetic wound healing, suggesting their capability as an effective antimicrobial tissue adhesive [34].

### 3.3. Nanocomposites or Composite Nanoparticles

Artificial synthesis of NPs is limited due to the fact that it involves high costs, energy consumption and additional resources to dispose of toxic by-products. Therefore, non-polluting, cheap and renewable sources such as plant extracts have gained popularity as they contain alkaloids, phenols, amino acids and proteins which are employed in the reduction and stabilization of Ag ions in AgNPs. For example, phenolic compounds from *Prosopis juliflora* have been used in AgNPs synthesis, reducing Ag^+^ [35]. Sood et al. employed *Ocimum sanctum* and incorporated optimized AgNPs into a carbopol gel base which achieved 96.20% wound closure by the 14th day [36]. In addition, it also presented an antibacterial inhibitory efficiency similar to commercial products against *Staphylococcus aureus* and *Pseudomonas aeruginosa* [36].

Gelatin is a natural, biodegradable, highly biocompatible, non-immunogenic polymer, rich in hydroxyproline that can reduce AgNO_3_ in situ through the hydroxyl groups. Moreover, its non-polar aminoacids help stabilize AgNPs. As a result, Ye et al. developed gelatin-chitosan-Ag, a new composite with a high density of pores with diameters of 100–250 µm, in which AgNPs were initially mixed with chitosan, cross-linked with tannic acid and cryodesiccated. The compound exhibited wound healing and anti-bacterial properties, while maintaining low cytotoxicity [37]. In addition, Shao et al. used *Barleria gibsoni* aqueous leaves to synthetize a ZnONPs gel efficient in treating both Gram-positive and Gram-negative infected burns [38].

Polymeric nanomaterial therapy employs polymeric nanoparticles (e.g., chitosan) as wound dressings or as delivery vectors due to the fact that they exhibit antibacterial and re-epithelialization properties [7]. Biopolymers are biocompatible polymeric networks that can absorb significant quantities of liquid, while also ensuring a moist wound environment [39].

Chitosan is a hydrophilic biopolymer commonly extracted from the exoskeleton of crustacean arthropods, characterized by high bioavailability and low toxicity, in addition to being one of the few polymers with antibacterial properties [39]. Chemically, this biopolymer is a linear polysaccharide containing D-glucosamine and N-acetyl glucosamine units. Apart from its affinity to metals, proteins and dyes, chitosan can also form complexes with anions, such as sulfate and phosphate, due to its cationic nature [40]. Moreover, its degradation products can stimulate synthesis of extracellular matrix components [39].

Various forms of chitosan have been investigated for wound-healing therapies, such as hydrogels, membranes, films, sponges and scaffolds [40]. Chitosan nanoparticles are characterized by superior permeability, antibacterial properties and immune modulation, while also being non-toxic to human cells. Chen et al. fabricated a novel acellular porcine dermal matrix in two steps. First of all, the acellular network was cross-linked with a naturally-derived oxidized chitosan oligosaccharide in order to obtain superior physicochemical traits [41]. Secondly, residual aldehyde groups within the newly formed compound were employed alongside Ag ions in order to generate AgNPs in situ (redox reaction) [41]. As a result, the generated scaffold was highly biocompatible and it exhibited wide-spectrum antimicrobial properties against *Escherichia coli* and *Staphylococcus aureus*, as well as accelerated wound healing [41]. Moreover, Hajji et al. reported that chitosan-polyvinyl alcohol-silver nanoparticles presented superior antioxidant and antimicrobial properties against both Gram-positive (*Staphylococcus aureus*) and Gram negative (*Escherichia coli* and *Klebsiella penumoniae*) bacteria than simple chitosan polymer. In addition, this complex significantly improved wound closure in vivo by inducing granulation and re-epithelialization, while exhibiting reduced cytotoxicity [42]. Holban et al. observed that coatings containing polylactic acid-chitosan-magnetite-eugenol nanospheres inhibited biofilm synthesis of *Staphylococcus aureus* and *Pseudomonas aeruginosa*, while also stimulating endothelial proliferation [43].

Gao et al. developed a thermo-sensitive hydrogel-based drug reservoir capable of releasing active substances on demand, triggered by near-infrared light stimulation. They mixed ciprofloxacin-loaded polydopamine nanoparticles and glycol chitosan, combining complementary antibacterial defense mechanisms. First of all, the positive charge of glycol chitosan attracted and trapped bacteria on the surface of the gel. Secondly, polydopamine nanoparticles can be activated by near-infrared light irradiation, leading to a photothermal response which generated local hyperthermia, thus inducing bacterial destruction. Moreover, polydopamine nanoparticles were loaded with ciprofloxacin which was released on demand through near-infrared light stimulation, even though the hydrogel complex exhibited minimal leakage under physiological conditions [44].

Hernandez Martinez et al. described a nanocomposite of AuNPs, chitosan and calreticulin for the therapy of diabetic lesions. Calreticulin is a 46 kDa calcium-binding protein found in the endoplasmic reticulum which regulates cytosolic and endoplasmic reticulum calcium levels. In addition, calreticulin acts as a molecular chaperone that controls the proper folding of proteins. The gold nanoparticles-chitosan-calreticulin composite promoted the growth, migration and differentiation of keratinocytes, fibroblasts and endothelial cells, without affecting cell viability, in vitro as well as in vivo. Moreover, the histological examination described granulation, re-epithelization and increased collagen synthesis [45].

Sun et al. encapsulated cathelicidin (antimicrobial peptide) extracted from king cobra in carboxymethyl chitosan nanoparticles and evaluated their effects on wound healing. Cathelicidin exhibits potent antibacterial properties in the presence of 1% NaCl. Moreover, it regulates cytokine release, increasing the expression of anti-inflammatory IL10, as well as downregulating pro-inflammatory cytokines, such as TNF-α, IL-1β and IL-6. Even though free cathelicidin is enzymatically degraded in less than 4 h, the combined nanomaterial overcomes this disadvantage through the constant release of free cathelicidin, which maintains active levels for longer periods of time, even in deeper skin layers. Cathelicidin-carboxymethyl chitosan nanoparticles only enhanced cell migration, without influencing keratinocyte growth. Moreover, in later stages of wound closure, the ratio of collagens I and III favored non-scaring healing [46].

Another biopolymer used in wound healing is cellulose, the main component of plant cell wall, thus making it extremely accessible. Structurally, it is composed of β-1,4 linked D-glucose units which form cellobiose repeating units. Cellulose accelerates wound closure through multiple growth factors such as epidermal growth factor and basic fibroblast growth factor that are released locally [21]. Nanocellulose has been employed in dressings due to its anti-infectious properties and the increased tensile properties of its scaffolds [47].

Bacterial cellulose is a biopolymer synthetized by various microorganisms, such as *Acetobacter*, *Pseudomonas*, *Salmonella*, through carbohydrate fermentations. Bacterial cellulose represents an excellent skin substitute due to its physico-chemical traits, such as very high surface area per unit, increased biocompatibility, hydrophilicity and nontoxicity. Bacterial nanocellulose is characterized by three-dimensional porous networks capable of retaining a high quantity of water, thus ensuring a damp environment, proper for wound healing [48]. Khalid et al. analyzed the healing properties of bacterial cellulose combined with ZnONPs and discovered that the added ZnONPs granted antibacterial properties [49]. Moniri et al. combined bacterial nanocellulose obtained from the fermentation of Gram-negative *Gluconacetobacter xylinus*, with silver nanoparticles (BNC-Ag). The nanocomposite improved wound healing and reduced *Staphylococcus aureus* colonization in vitro [50].

#### Nano-Carriers for Wound Healing

Nanomaterials can also act as carriers for therapeutic agents, controlling their release. Nitric oxide exhibits a crucial role in inflammatory pathways, cellular growth, angiogenesis and extracellular matrix deposition and remodeling. In addition, nitric oxide exhibits wide spectrum antibacterial properties, while also interfering with biofilm synthesis. As a result, many studies have tried to produce a delivery system with high loading capacity, controlled release and reduced cytotoxicity [7]. Nitric oxide-releasing poly(lactic-co-glycolic acid)-polyethylenimine nanoparticles were designed to provide prolonged nitric oxide release, antibacterial efficacy against methicillin-resistant *Staphylococcus aureus* and *Pseudomonas aeruginosa*, as well as accelerated wound closure in vivo [51].

Wang et al. developed a novel pH-responsive calcium alginate hydrogel laden with protamine nanoparticles and hyaluronan oligosaccharides for diabetic wound therapy. The principle was that as the hydrogel absorbed wound exudates, the alkaline fluids would decrease, thus triggering a pH shift which determined the calcium alginate to release active protamine nanoparticles, destroying both Gram-positive and Gram-negative bacteria. Moreover, hyaluronan oligosaccharides stimulated endothelial cell proliferation, motility and angiogenesis, through enhanced expression of vascular endothelial growth factor [52].

Curcumin, the dynamic component of the nutritional spice turmeric, also presented interest due to its antibiotic and antioxidant properties, as well as stimulating the production of granulation tissue. A recent study attempted to increase curcumin’s poor solubility by encapsulating it into a saline-hydrogel nanoparticle vector, and the results demonstrated in vitro efficacy against methicillin-resistant *Staphylococcus aureus* and enhanced wound closure activity [53]. Ghaffari et al. stated that both gel and ointment preparations containing curcumin-Ampicillin solid lipid nanoparticles exhibited antibacterial effects, while also increasing wound closure rate compared to control and placebo groups, both in vitro and in vivo, without additional toxicity [54]. In addition, in a mouse model experiment, Moradi et al. studied the associated effect of pulse photo-bio-modulation with curcumin loaded super-paramagnetic iron oxide nanoparticles and reported accelerated wound healing and an important reduction in *Staphylococcus aureus* colonies [55].

### 3.4. Coatings and Scaffolds

Another application of nanomaterials is the production of scaffolds that mimic the properties of extracellular matrix. Several techniques are used to produce them, such as electro-spinning, self-assembly and phase separation. The most employed is electro-spinning which creates porous polymeric nanofibers that can be used as hybrid scaffolds for the adherence and development of fibroblasts in wounds [7]. Other nano-polymers, such as dendrimers, can exhibit anti-inflammatory properties, while also forming networks that can incorporate antibacterial agents (e.g., silver) [56].

Using a porcine model of superficial partial thickness wounds, Haik et al. studied the healing properties of a polymer nanofiber dressing applied by a mobile handheld electrospinning device. Regarding safety (risk of infection, delayed healing) the results were similar to the traditional paraffin tulle gras dressing. The non-touch technique and the ease of application represent advantages of this therapeutic option [57].

Electrospun chitosan-poly-vinyl alcohol nanofibrous blend scaffolds were applied to treat models of diabetic wounds in rats, showing increased healing rates compared with controls [58].

The in vivo study in Wistar rats of a electrospun nanofiber membrane enhanced with silver nanoparticles showed multiple favorable effects: low cytotoxicity, long-term antibacterial activity, reduced inflammation and enhanced wound healing [59].

An electrospun composite scaffold with curcumin showed low cytotoxicity and anti-oxidant effect in vitro, as well as enhanced wound healing in vivo, in a full-thickness dermal model for Wistar rats [60].

Other nanoscale system compounds attempted to deliver growth factors, such as recombinant human epidermal growth factor, in order to stimulate the healing of full-thickness diabetic wounds. However, their application was limited by the highly proteolytic environment, as well as the down-regulation of the associated growth factor receptors and signaling molecules in chronic lesions [61]. In contrast, other authors described better results when using basic fibroblast growth factor (bFGF). For example, Zhang et al. described a hydrogel that employed Ca^2+^ as a cross-linker between carboxyl groups (–COO) and that was capable of releasing preloaded bFGF. They observed that both calcium and bFGF stimulated fibroblast growth and differentiation, and that the effect was more potent in the early phases of re-epithelialization, inducing wound contraction, both in vitro and in vivo. In addition the hydrogel presented two profiles of bFGF release: an initial burst which accounted for approximately 40% in the first 12 h and a secondary, prolonged phase, which ensured a constant release of bFGF over 8 days. These dynamics suggest that the compound achieves increased levels and stability of local bFGF over a longer period of time [62].

Furthermore, electro-spun nano-fibrous meshes have been used as networks for gene encapsulation in wound-dressing materials. Gene-activated matrix therapy, which associates gene therapy and tissue engineering, has the advantage of increasing or decreasing the expression of a target gene involved in regeneration. DNA is also more stable compared to growth factors. However, more studies need to be performed in order to obtain additional data about gene therapy effectiveness, bioavailability and safety in the presence of wound bacterial colonization [7].

Wang et al. optimized a novel gene delivery system based on cathelicidin (antimicrobial peptide LL-37) grafted on ultra-small gold nanoparticles for the topical treatment of diabetic lesions. This system combines the advantages of cationic AuNPs with those of cathelicidin, increasing the overall antibacterial activity compared to the free antimicrobial peptide and the AuNPs alone. Additionally, the LL37-AuNPs complex enhanced cellular and nucleus penetration, thus achieving high gene delivery efficiency. Furthermore, this combined system characterized by high biocompatibility, promoted angiogenesis through increased VEGF expression, and also improved re-epithelialization and granulation tissue formation, both in vitro and in vivo [63].

Zgheib et al. conjugated microRNA-146a (miR-146a) with cerium oxide nanoparticles to enhance the diabetic wound healing process. MicroRNAs have been demonstrated to control the transcription of certain genes, thus regulating pro-inflammatory cytokine synthesis [64]. For example, miR-146a represses the production of interleukin-1 receptor-associated kinase 1 (IRAK1) and tumor necrosis factor receptor-associated factor 6 (TRAF6). IRAK1 and TRAF6 are crucial regulators of the NFkB pathway (pro-inflammatory nuclear transcription factor kB pathway), increasing NFkB activity, thus overexpressing the genes encoding IL-6 and IL-8. It has been observed that, in diabetic wounds, the expression of miR-146a is decreased, which accounts for the lack of suppression of IRAK1 and TRAF6 synthesis. In turn, NFkB pathway is stimulated in excess, leading to the overexpression of the pro-inflammatory cytokines IL-6 and IL-8 [65].

In addition, cerium oxide nanoparticles have the ability to scavenge reactive oxygen species (ROS) due to the valence structure of the cerium atom in which 2 oxidation states (3^+^ and 4^+^) coexist, depending on the presence of ROS. Therefore, these nanoparticles may reduce oxidative stress and restore the balance between oxidants and antioxidant enzymes in diabetic lesions. The results showed that a 100-ng dose of cerium oxide nanoparticles-miR-146a improved diabetic wound healing without affecting wound strength or elasticity [64].

Stem cell research represents another possible beneficiary of the nanoscaffold technology, due to the fact that these nano-scale networks could aid stem cell migration and differentiation, in order to promote re-epithelialization and angiogenesis. However, more studies have to be conducted in order to precisely predict the dynamics of stem cells during wound healing [66].

The multiple properties of nanomaterials regarding antimicrobial effects, the stimulation of wound healing by inducing angiogenesis, re-epithelialization, granulation and/or collagen synthesis, as well as their role in the prevention of abnormal scarring represent advantages compared to standard treatment and arguments towards their use in clinical practice (Table 1).

Further investigations are needed in order to establish a role in the prevention of wound development or progression.

## 4. Conclusions

Wound healing is an intricate three-staged process involving inflammation, proliferation and remodeling. The physiology of the healing process may be perturbed by both internal and external factors, the most important being contamination with microorganisms. In order to achieve a good recovery with minimal scarring, researchers have tried to develop many topical products that can provide a moist environment, as well as antibacterial activity. However, chronic wounds still remain a challenge due to multidrug-resistant microorganisms and bacteria biofilms. Nanoparticles, due to their superior surface-to-volume ratio, can be efficiently employed in countless medical applications, including wound therapy. Metal nanoparticles such as silver, gold and zinc possess outstanding properties such as low toxicity and antibacterial activity, making them perfect candidates for integration in wound dressings. Moreover, nanocompounds which encapsulate growth factors, genes or stem cells are currently under development, thus offering new treatment modalities in the near future. More studies have to be conducted in order to uncover the entire potential, as well as the risks associated with the usage of nanomaterials.

## Figures and Tables

**Figure 1 materials-12-02176-f001:**
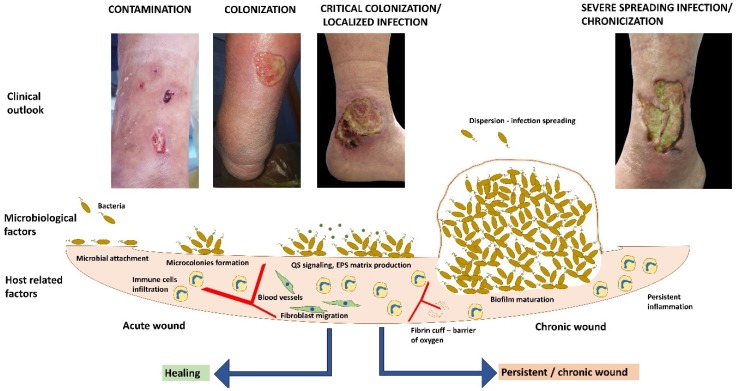
Correlation between the clinical outlook, microbiological and host-related factors in acute and chronic wounds.

**Figure 2 materials-12-02176-f002:**
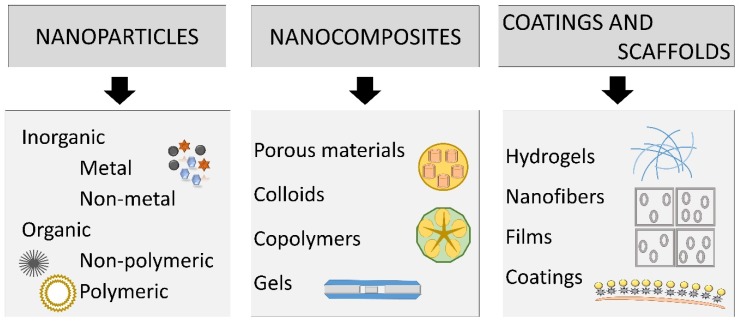
Main types of nanomaterials which can be used for wound treatment.

**Table 1 materials-12-02176-t001:** Properties of nanomaterials developed for wound management. bFGF—basic fibroblast growth factor.

Material		Properties			References
		Antibacterial	Stimulation of Wound Healing ^1^	Prevention of Abnormal Scarring	
Nanoparticles	Silver nanoparticles	+	+	+	[7,21,24,25]
	Gold nanoparticles	+	+	Unknown	[10,21,30]
	Zinc oxide nanoparticles	+	+	Unknown	[7,16,21,33]
	Composite nanoparticles Eg. chitosan nanoparticles, nanocellulose	+	+	+	[7,36,37,39,41,42,43,44,45,46,47,49,50]
Nanocarriers	Eg. Nitric oxide-releasing poly(lactic-co-glycolic acid)-polyethylenimine nanoparticles, curcumin loaded super-paramagnetic iron oxide	+	+	Unknown	[51,52,53,54,55]
Coatings and Scaffolds	Eg. hydrogel preloaded with bFGF, Hydrogels reinforced with AgNPs	++	++	Unknown	[63,65,66]

^1^ (angiogenesis, re-epithelialization, granulation and/or collagen synthesis).

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
