# Peer review of "Nanomaterials for Wound Healing and Infection Control"

_materials, 2019, doi:10.3390/ma12132176_

Reviewer 1 Report

I would like to add some comments on the chemical-physical part related to the formation of nanoparticles and the possible toxicological aspect

Author Response

Dear reviewer,

Many thanks for your time and valuable comments.

Please find below step-by-step reply to your comments:

”I would like to add some comments on the chemical-physical part related to the formation of nanoparticles and the possible toxicological aspect”

Answer: Details regarding the physical and chemical properties of nanoparticles and their impact were provided.

Kind regards,

The authors

Reviewer 2 Report

Mara Madalina Mihai, Monica Beatrice Dima, Bogdan Dima and Alina Maria Holban reported a review manuscript entitled, “Nanomaterials for wound infection control” to Materials.

 “Introduction”, three phases are overlapped, and this should be clarified.

“Wound Therapy” section seems redundant and can be deleted in a current form, but If to include, it should be compared to and referred to subsequent sections of “nano-particles” and “nano-carriers”

In growth factors, bFGF, basic fibroblast growth factor, should be included and discussed.

All the nanomaterials should be divided into “treatment” and “prevention and management” and further application to each organism should be elucidated.

Author Response

Dear reviewer,

Many thanks for your time and valuable comments.

Please find below step-by-step reply to your comments:

1.     “Introduction”, three phases are overlapped, and this should be clarified.

Answer: We tried to clarify the process of wound healing. ‘Wound healing involves three biological stages: inflammation, which usually lasts up to 6 days, proliferation, which typically covers the following 2 weeks, and remodeling which continues for up to 2 years. It is important to mention that in vivo the phases overlap due to several anatomical and physiological factors such as the constant intercellular signaling that implies real-time feedback control of pro-inflammatory and anti-inflammatory cytokine release’

2.     “Wound Therapy” section seems redundant and can be deleted in a current form, but If to include, it should be compared to and referred to subsequent sections of “nano-particles” and “nano-carriers”

Answer:  ‘Wound therapy’ section was updated (Current Wound therapy) and refers to commonly used wound dressings and more conventional treatment options (describing their advantages and drawbacks), setting the stage for the subsequent sections which cover a modern approach, via nanoparticles, of wound management.

3.     In growth factors, bFGF, basic fibroblast growth factor, should be included and discussed.

Answer: Thank you for your suggestion. We added a paragraph in the ‘Coatings and scaffolds’ section discussing the effects of bFGF. Growth factors involved in wound healing represent an appealing and complex subject that could also be treated on its own in extenso.

4.     All the nanomaterials should be divided into “treatment” and “prevention and management” and further application to each organism should be elucidated.”

Answer: We tried to summarize the properties of nanomaterials in Table 1. All their properties (antibacterial, stimulation of wound healing, prevention of abnormal wounds) can represent advantages in the treatment of chronic wounds. Since these are treatments that are not yet used in clinical practice we did not find relevant data on their indications for the prevention of development of chronic wounds or the prevention of infection, very interesting and useful subjects that could be investigated in further studies.

Table 1. Properties of nanomaterials. bFGF- basic fibroblast growth factor.

Kind regards,

The authors

Reviewer 3 Report

I think the manuscript would be worth of publishing in the this Journal providing following remarks/suggestions will be accounted for. I hope that they may help improving the manuscript appropriately. Alternatively, the authors should give reasonable comments and answers.
1.The results are to few  discussed. A more deeply discussion of the results is needed.
2.In the introduction, the authors have explained the scientific significance of  nanoparticles, chemistry, and application, but it could be more attractive to readers if the authors include wound healing using  nanoparticles that is need of the hour for many nanomaterials.
3.Authors can consider offer difference  nanoparticles  this SEM or TEM  analysis

Author Response

Dear reviewer,

Many thanks for your time and valuable comments.

Please find below step-by-step reply to your comments:

1.”I think the manuscript would be worth of publishing in this Journal providing following remarks/suggestions will be accounted for. I hope that they may help improving the manuscript appropriately. Alternatively, the authors should give reasonable comments and answers. The results are too few discussed. A more deeply discussion of the results is needed.”

Answer: Thank you for the appreciation. We tried to expand discussions.

 2. “In the introduction, the authors have explained the scientific significance of nanoparticles, chemistry, and application, but it could be more attractive to readers if the authors include wound healing using nanoparticles that is need of the hour for many nanomaterials.”

Answer: We have briefly presented the effects of each nanomaterial in wound healing as we have presented them. Also, their physico-chemical properties were discussed.

3.”Authors can consider offer difference nanoparticles  this SEM or TEM  analysis”.

Answer: Thank you for the interesting suggestion – we added a reference to SEM and TEM when discussing about the configuration of nanomaterials.

Kind regards,

The authors

Round  2

Reviewer 2 Report

Mara Madalina Mihai, Monica Beatrice Dima, Bogdan Dima and Alina Maria Holban reported a review manuscript entitled, “Nanomaterials for wound healing and infection control” to Materials.

Dividing nano-materials into nanoparticles, nanocomposites and coating and scaffolds and combining two or three, such as nanoparticle and nanocomposites or nanoparticles, nanocomposites and coating and scaffolds should be presented.

Each “nanomaterials” should accompany more physiological functional data, preferably with in vivo or clinical data.

Too much limiting to the “Materials” may alter the meaning to the value of these properties.

Author Response

„Dividing nano-materials into nanoparticles, nanocomposites and coating and scaffolds and combining two or three, such as nanoparticle and nanocomposites or nanoparticles, nanocomposites and coating and scaffolds should be presented.

Each “nanomaterials” should accompany more physiological functional data, preferably with in vivo or clinical data.

Too much limiting to the “Materials” may alter the meaning to the value of these properties.”

Answer: Dear Reviewer, many thanks for your recommendations, they helped us improve our paper. We have included more recent data regarding clinical trials and in vivo applications of some nanoparticles used for wound therapy (especially silver and zinc oxide).

Also, we have included recent examples of some new composite materials and combinations between nanoparticles and coatings or scaffolds in subsection 3.4. Moreover, we have highlighted the progress made in vivo (and also in vitro) in all sections discussing various nanomaterials (section 3) and inserted a table showing the main types of nanomaterials and their properties in wound management.